# A Power Based Analysis for a Transonic Transport Aircraft Configuration through 3D RANS Simulations

**Peijian Lv** **, Defu Lin and Li Mo \***

School of Aerospace Engineering, Beijing Institute of Technology, Beijing 100811, China
\* Correspondence: levin_mott@163.com; Tel.: +86-13911436237

**Abstract:** This paper presents a power-based analysis through 3D Reynolds-averaged Navier–Stokes simulations for a typical transonic transport aircraft resented by the DLR-F6 model. Two configurations were employed in CFD simulations. The original F6 model geometry was defined as the wing body configuration, and a wake-filling actuator disc was added to the F6 model to establish the BLI configuration. This study proposes a segregated 3D computational domain in RANS simulations to track the change in power terms in the flow field so that the power conversion process can be studied and visualized. For the wing body configuration, the power-based analysis illustrated the power conversion process, showing that about 35% of the total power input remains in the form of the mechanical power of aircraft wake at the outlet plane. For the BLI configuration, 22% of the total power input was left in the form of the mechanical power of downstream flow mixed with the wake and jet at the outlet plane. This study elaborates on the error of the mechanical power imbalance, showing that the convergence in aircraft drag does not necessarily lead to a small error in 3D RANS simulations. The high value of power imbalance error is associated with the wing.

**Keywords:** power-based analysis; power imbalance error; 3D RANS simulation

## 1. Introduction

Boundary layer ingestion (BLI) has been studied as a promising technology that utilizes favorable airframe propulsion integration to reduce aircraft fuel consumption. BLI has been adopted by novel aircraft concepts, such as Boeing blended wing body aircraft [1], silent aircraft [2], MIT D8 aircraft [3,4], the "propulsive fuselage" concept [5], H2020 CENTERLINE aircraft [6], and NASA STARC-ABL aircraft [7,8]. The benefits due to BLI range from 3% to 10% for these aircraft concepts. The flow mechanisms of power saving due to BLI were recognized and explained by Betz [9] in the early days. The flow mechanisms of BLI have become clear since the introduction of power-based methods, which enabled the quantitative study of the actual power consumptions of aircraft using BLI. Drela [10] introduced a power balance method to evaluate mechanical power in the flow field, showing that the power consumption associated with the aircraft wake and jet is eliminated by BLI, as shown in Figure 1. Arntz et al. [11,12] established an exergy-based method to include thermal power in the study of BLI. Lv et al. [13,14] utilized a mechanical power analysis to show that a BLI propulsor utilizes the power of the ingested boundary layer flow as input power and reduces the wasted power in the downstream flow. These power-based methods are different from a traditional method that breaks down the aerodynamic force imposed on aircraft surfaces [15]. Power-based methods evaluate the power in the flow field around the aircraft so that the actual power consumption of the aircraft is calculated without ambiguity [16].

Recent efforts were made to inspect the potential benefits of BLI in detailed aircraft design. Computational fluid dynamics (CFD) simulation has been widely used as a tool to design aircraft shapes. Power-based methods are adopted to access the power terms in the flow field in the postprocessing of simulations. Elmiligui et al. [17] performed an

analysis of axisymmetric fuselage–propulsor configurations to evaluate the power savings due to BLI. Sanders and Laskaridis [18] employed Reynolds-averaged Navier–Stokes (RANS) simulations to study the potential benefit of BLI with a 2D axisymmetric fuselage geometry. Baskaran et al. [19] performed 2D CFD simulations to optimize the shape of an axisymmetric fuselage using BLI in transonic flight conditions. Besides the aforementioned studies using 2D simulations, 3D flow simulations are employed in the study of BLI. Blumenthal et al. [20] presented a study to optimize fuselage geometry based on a common 3D research model to evaluate the benefit of BLI. Kenway and Kiris [21] addressed the flow distortion issue of BLI by optimizing fuselage geometry through 3D RANS simulation. Gray et al. [22] employed 3D RANS analysis to perform the optimization of a coupled aeropropulsive aircraft model with BLI.

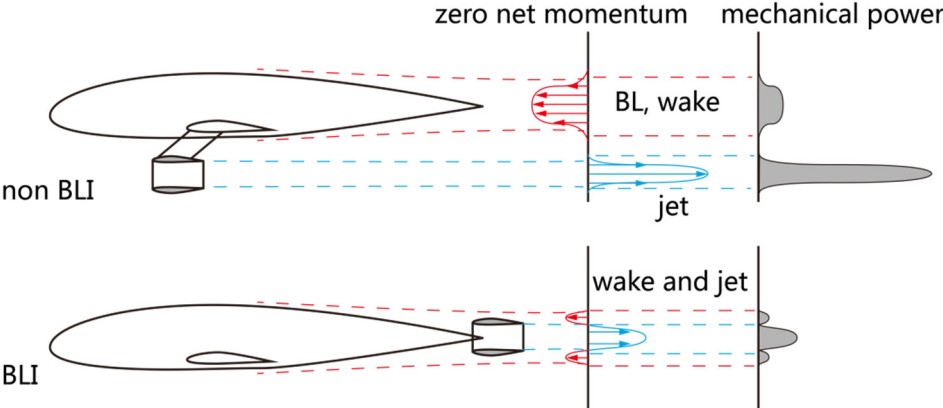

**Figure 1.** Boundary layer ingestion benefits of reducing the mechanical energy of the downstream wake and jet (adapted from reference [3]).

Previous works using power-based methods combined with CFD simulations were based on a well-defined control volume so that the power terms of the entire flow field could be evaluated [17–20]. Nevertheless, the simplicity of control volume was not capable of studying the detailed process of power conversion in these previous works. This limitation prevents the investigation of how aircraft components influence the power conversion process. This incapability suggests further development of power-based analysis. To deal with this issue, this study introduces a segregated 3D computational domain aiming to trace the changing process of power terms in the flow field over a typical transonic transport aircraft. With this improved power-based analysis, to the knowledge of the authors, this is the first time that the detailed process of power conversion in a 3D simulated flow field over an aircraft has been examined and visualized.

Based on the aforementioned framework, this research makes efforts to examine the power conversion process in the flow field over a transonic transport aircraft through 3D compressible RANS simulations as an extension of the previous work of studying power conversion in 2D incompressible flow fields [23]. The well-known DLR-F6 model was selected as the baseline configuration to study the power conversion process in the flow field over a typical aircraft airframe [24–27]. The BLI configuration was established by combining the F6 model with a tail-mounted wake-filling actuator disc model. This wake-filling actuator disc model mimics an ideal BLI propulsor to re-energize the ingested boundary layer flow into the state of free stream so that the mechanical energy of the downstream wake and jet is kept to a minimum. To evaluate how well mechanical conservation is satisfied in simulations, this study introduced a power imbalance error study. Possible impact factors influencing the power imbalance error were examined. The power conversion process of the baseline wing body configuration and BLI configuration were analyzed and are discussed.

This study is organized as follows: Section 2 introduces the method of power-based analysis used in the 3D simulations and the segregated computational domain of aircraft.

Section 3 presents the simulation results, and the power imbalance error of 3D RANS simulations is analyzed. The power conversion process of the aircraft is presented and discussed. Conclusions are provided in Section 4.

## 2. Methodology

### 2.1. Power-Based Analysis in 3D RANS Simulations

Computational fluid dynamics simulations obtain the flow field so that the continuity, momentum, and energy equations are satisfied. On the other hand, the mechanical energy equation is different from the energy equation. The former is obtained by multiplying the momentum equation with velocity [28], while the latter is related to the first law of thermodynamics. The diagram in Figure 2 illustrates the relation between the conservations of mechanical energy (mechanical energy/Navier–Stokes momentum equation) and energy (Navier–Stokes energy equation). Power-based analysis utilizes the integral form of the kinetic energy equation. As a result, numerical simulations provide sufficient and necessary conditions for power-based analysis due to the conservation of momentum.

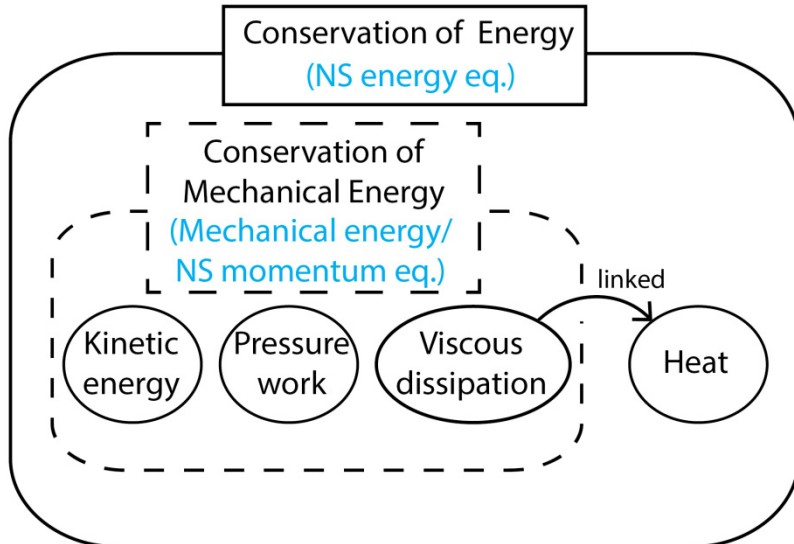

**Figure 2.** The diagram of the conservation of mechanical energy and total energy [23].

In power-based analysis, the integral mechanical energy equation can be simplified as Equation (1). This equation establishes the equilibrium of mechanical power by using three terms, namely power inputs; outputs; and an error, $e$. The power inputs and outputs are introduced in the following paragraphs. The error, $e$, is elaborated in detail in the section on power imbalance error.

$$P_{in} = P_{out} + e \qquad (1)$$

Figure 3 depicts a control volume of a simplified transport aircraft that is bounded by an inlet, an outlet, a cylindrical far-field boundary, and the aircraft surface. A survey plane is introduced as an internal plane perpendicular to the flow direction. The aircraft surface consists of a fuselage, wing, and propulsor. These aircraft components can be categorized into two groups: the fuselage and wing are power-consuming components, while the propulsor is the power-adding component. The power terms associated with the aircraft components and control volume are listed in Table 1. For the entire control volume, the total power input is the summation of the wake energy inflow (rate), $\dot{E}_{w,in}$; drag power, $DV_\infty$; and shaft power (mechanical power addition), $P_s$, as expressed by Equation (2). The total power output is the summation of the wake energy outflow (rate), $\dot{E}_{w,out}$; thrust power, $TV_\infty$; and viscous dissipation rate, $\Phi$, as given in Equation (3). The individual power terms are elaborated in the following paragraphs. The power balance equation of the entire

control volume can be established by combining these power terms and the error, *e*, as given in Equation (1).

$$P_{in} = \dot{E}_{w,in} + DV_{\infty+}P_S \tag{2}$$

$$P_{out} = \dot{E}_{w,out} + TV_{\infty} + \Phi \tag{3}$$

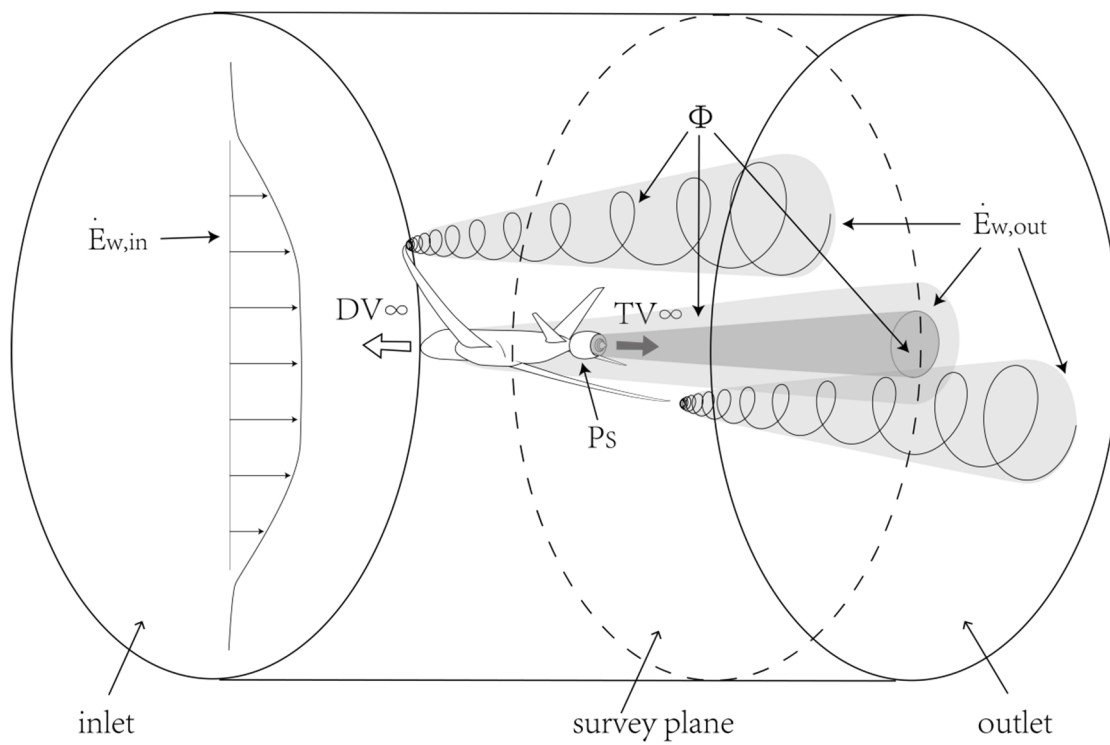

**Figure 3.** Control volume of a simplified aircraft with a BLI propulsor.

**Table 1.** Power terms in the control volume of an aircraft.

|  | Control Volume | Wing/Body | Propulsor |
|---|---|---|---|
| Input term | $\dot{E}_{w,in}$ | $DV_{\infty}$ | PS |
| Output term | $\dot{E}_{w,out}$, $\Phi$ | - | $TV_{\infty}$ |

The power term of the wake energy flow rate, $\dot{E}_w$, refers to the mechanical energy passing through a plane. For the inlet plane, it corresponds to the mechanical energy inflow, $\dot{E}_{w,in}$. Its value is simply zero for the free stream incoming flow, $V_{\infty}$, but it has a finite value for a possible headwind or backwind entering the control volume. For the out plane, $\dot{E}_{w,out}$, it is the mechanical energy of the aircraft wake and jet that flows out of the control volume. For an internal survey plane (SP), the surface integral of $\dot{E}_w$ evaluates the mechanical energy of the flow through the plane. On the other hand, depending on the physical property, $\dot{E}_w$ can be broken down into the kinetic energy flow rate, $K\dot{E}_w$, and the pressure power, $\dot{E}_p$, as shown in Equation (4). The former refers to the kinetic energy of the flow, as given in Equation (5). The latter is the pressure work, which might transform isentropically into $K\dot{E}_w$, as expressed by Equation (6). It is noted that $\dot{E}_w$ is zero for the free stream, and the magnitude of $\dot{E}_w$ is based on the state of the free stream flow. For the entire control volume, the inflow and outflow terms ($\dot{E}_{w,in}$ and $\dot{E}_{w,out}$) are listed in Table 1.

$$\dot{E}_w = K\dot{E}_w + \dot{E}_p \tag{4}$$

$$K\dot{E}_w = \dot{E}_a + \dot{E}_v = \iint\limits_{SP} \rho u \frac{1}{2}(u - V_{\infty})^2 dS + \iint\limits_{SP} \rho u \frac{1}{2}\left(v^2 + w^2\right) dS \tag{5}$$

$$\dot{E}_p = \iint\limits_{SP} (p - p_\infty)(u - V_\infty)dS \tag{6}$$

The drag force, D, is exerted on the surface of the aircraft components of the body, wing, empennage, etc. Regarding the conversion of mechanical power, the drag power, $DV_\infty$, is the input power of an aircraft. It corresponds to the power of an ideal force dragging the aircraft in a flow field. This power is obtained by simply multiplying D by $V_\infty$.

It is straightforward that the propulsive system converts shaft power, $P_s$, into thrust power, $TV_\infty$. Therefore, $P_s$ is an input term, while thrust power, $TV_\infty$, is an output term. In this study, $P_s$ is the mechanical power addition that enters the control volume through the boundary of the propulsor surface. For a simplified propulsor of actuator disc model only changing the axial flow, the expression of $P_s$ is given in Equation (7). The output power term thrust power, $TV_\infty$, is obtained by multiplying thrust, T, by $V_\infty$.

$$P_s = \oiint\limits_{prop} (p - p_\infty)\, udS = \oiint\limits_{prop} \Delta p\, udS \tag{7}$$

Viscous dissipation, $\Phi$, is an output term of the control volume. $\Phi$ exists in the viscous region (boundary layer and wake) where the velocity gradient exists. This term uses the volume integral of a domain. The integrand of $\Phi$ contains nine components. According to the physical property, it can be broken down into laminar, turbulent, and bulk viscous dissipation, as expressed by Equation (8). The expressions of the three components are given in Equations (9)–(11). The turbulent viscosity, $\mu_{tur}$, is introduced to access the turbulent dissipation, $\Phi_{tur}$, by applying the Boussinesq assumption. It is noted that $\Phi$ is a sink term in the mechanical energy equation.

$$\Phi = \Phi_{lam} + \Phi_{tur} + \Phi_{bulk} \tag{8}$$

$$\Phi_{lam} = \iiint\limits_{CV} \mu_{lam}\left[\left(\frac{\partial v}{\partial x} + \frac{\partial u}{\partial y}\right)^2 + \left(\frac{\partial w}{\partial y} + \frac{\partial v}{\partial z}\right)^2 + \left(\frac{\partial u}{\partial z} + \frac{\partial w}{\partial x}\right)^2\right]dV \tag{9}$$

$$\Phi_{tur} = \iiint\limits_{CV} \mu_{tur}\left[\left(\frac{\partial v}{\partial x} + \frac{\partial u}{\partial y}\right)^2 + \left(\frac{\partial w}{\partial y} + \frac{\partial v}{\partial z}\right)^2 + \left(\frac{\partial u}{\partial z} + \frac{\partial w}{\partial x}\right)^2\right]dV \tag{10}$$

$$\Phi_{bulk} = \iiint\limits_{CV} (\mu_{lam} + \mu_{tur})\left[2\left(\frac{\partial u}{\partial x}\right)^2 + 2\left(\frac{\partial v}{\partial y}\right)^2 + 2\left(\frac{\partial w}{\partial z}\right)^2\right]dV \tag{11}$$

### 2.2. Model Geometry and Flight Conditions

This study employs the DLR-F6 model, which has been extensively studied through numerical simulations and wind tunnel experiments. The results of these studies have been documented in detail [24,25,27,29]. The DLR-F6 was selected in this study as a well-established reference to examine the power-based analysis in the 3D flow field. The original DLR-F6 geometry is called the wing body configuration in the following discussions.

This study establishes a BLI configuration by adding an actuator disc model to the wing body configuration, aiming to ingest the boundary layer flow developed over the body surface. This actuator disc is 100 mm in diameter (67% of body diameter), located at the aft fuselage with the axial location of 1181 mm (0.99% of the body length), as shown in Figure 4. The specifications of the wing body configuration and the BLI configuration are listed in Table 2. For this DLR-F6 geometry, the test flight condition was subsonic with the Mach number of 0.75, and the flow conditions were kept the same as in the reference studies. The key parameters of the test conditions are listed in Table 3.

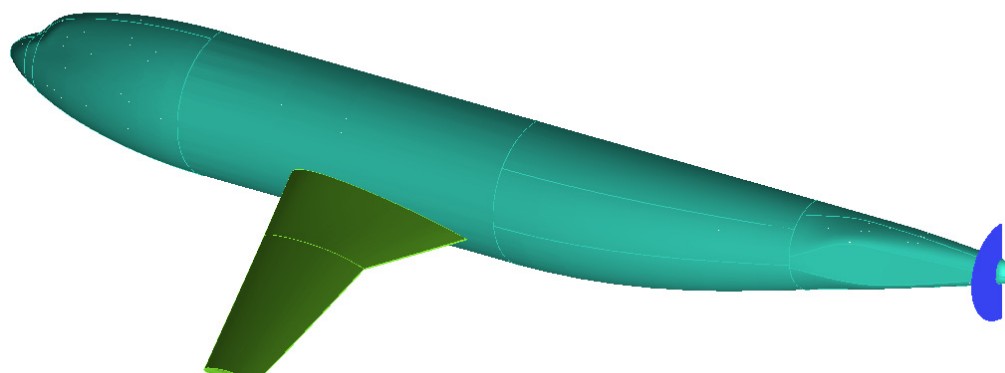

**Figure 4.** The geometry of the DLR-F6 wing body configuration and the BLI configuration (half model).

**Table 2.** Specifications of DLR-F6 geometry (body wing and BLI configurations).

| Parameters | Value |
|---|---|
| Mean aerodynamic chord | 141.2 mm |
| Projected half span | 585.6 mm |
| Half-model reference area, $S_{ref}$ | 72,700.0 mm$^2$ |
| Fuselage diameter | 148.4mm |
| The diameter of the actuator disc | 100 mm |
| X location of the actuator disc (from fuselage nose) | 1181 mm |

**Table 3.** Test condition of DLR-F6 model.

| Parameters | Value |
|---|---|
| Mach number | 0.75 |
| Air model | Perfect gas |
| Temp | 18 °C |
| Free stream velocity, $V_\infty$ | 260.4 m/s |

*2.3. Segregated Computational Domain and Simulation Conditions*

The computation domain is the key in power-based analysis. The domain includes the half model of DLR-F6 to reduce computational resources. The model locates at the center part of this domain, and it is encompassed by external boundaries, including an inlet, a symmetry plane, a cylinder boundary, and an outlet plane, as depicted in Figure 5. The external boundary, as the far-field boundary, is sufficiently large. According to reference [24], the far-field boundary is about 100 times the mean aerodynamic chord (MAC) away from the aircraft. The radius of the cylinder boundary is about 100 times the MAC, and the length of the domain is 130 times the MAC.

This computational domain was used in the two cases of the wing body configuration and the BLI configuration so that the influences of the mesh were eliminated when comparing the simulation results of the two cases. The switch between the two configurations could be realized by changing the boundary condition of the actuator disc plane: the plane was set as an internal plane for the wing body configuration and it could be changed into a pressure jump plane for the BLI configuration. As for the surfaces of wing and body, the boundary condition was set as the wall.

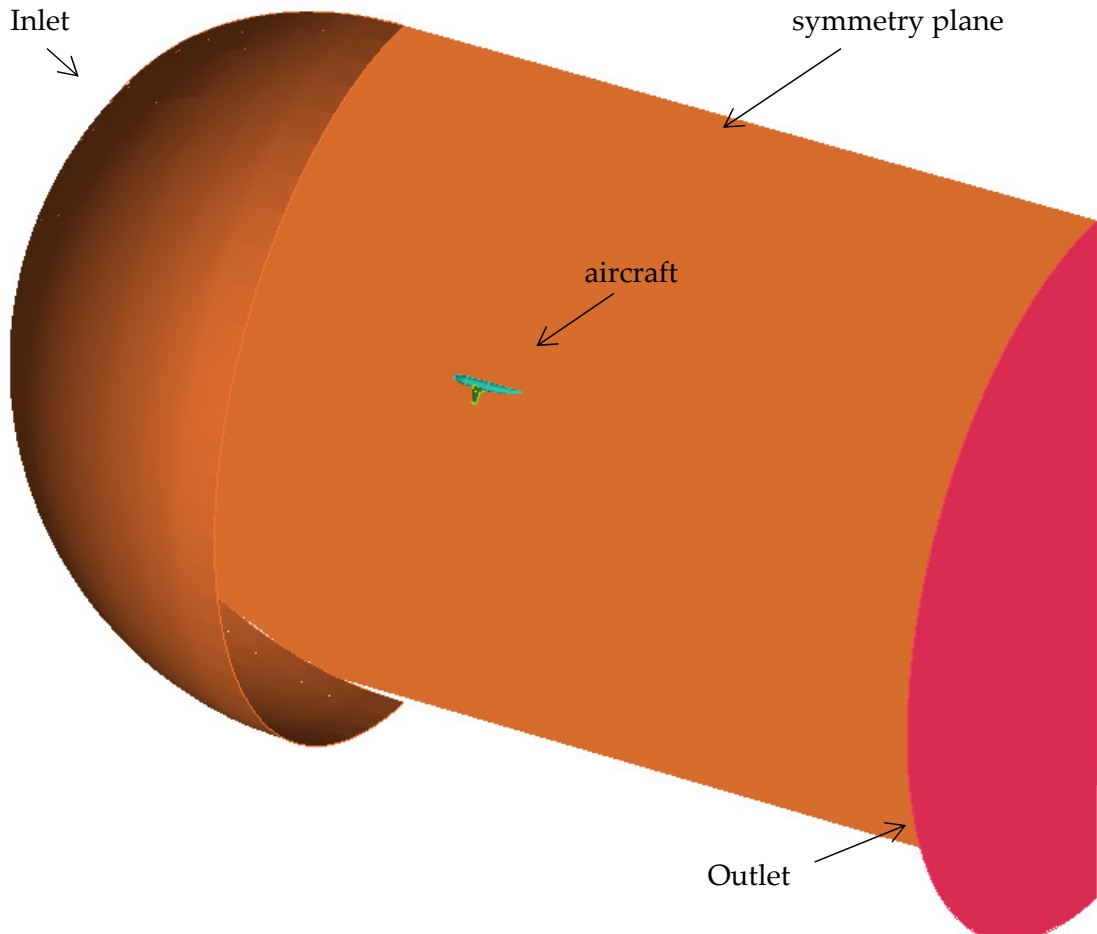

**Figure 5.** Boundaries of computation domain.

A key feature of this power-based analysis is the segregation of the computational domain. By dividing the entire domain into individual subdomains, the power terms for each subdomain can be evaluated and the change in power terms is tracked [23]. Moreover, the division of the domain can be associated with the geometry of aircraft components. The mechanical power conversion of an individual component could be examined, enabling us to study the impacts of a specific aircraft component. As for the entire aircraft, the big picture of power conversion was established by combining the power conversions of the subdomains. This study defines 13 internal survey planes that segregated the computational domain into 14 subdomains. These subdomains correspond to the components of wing, body, and propulsor, as illustrated in Figure 6.

To obtain the 3D flow field of the DLR-F6 model, this study employed ANSYS Fluent software to perform 3D compressible RANS simulations. The spatial discretization of simulations employed the second-order upwind and the third-order MUSCL methods. This study used the SIMPLE scheme for the pressure–velocity coupling. As for turbulent flow, three models were employed, namely the S-A model, the standard k-$\varepsilon$ model, and Mentor's k-$\omega$ SST model [30].

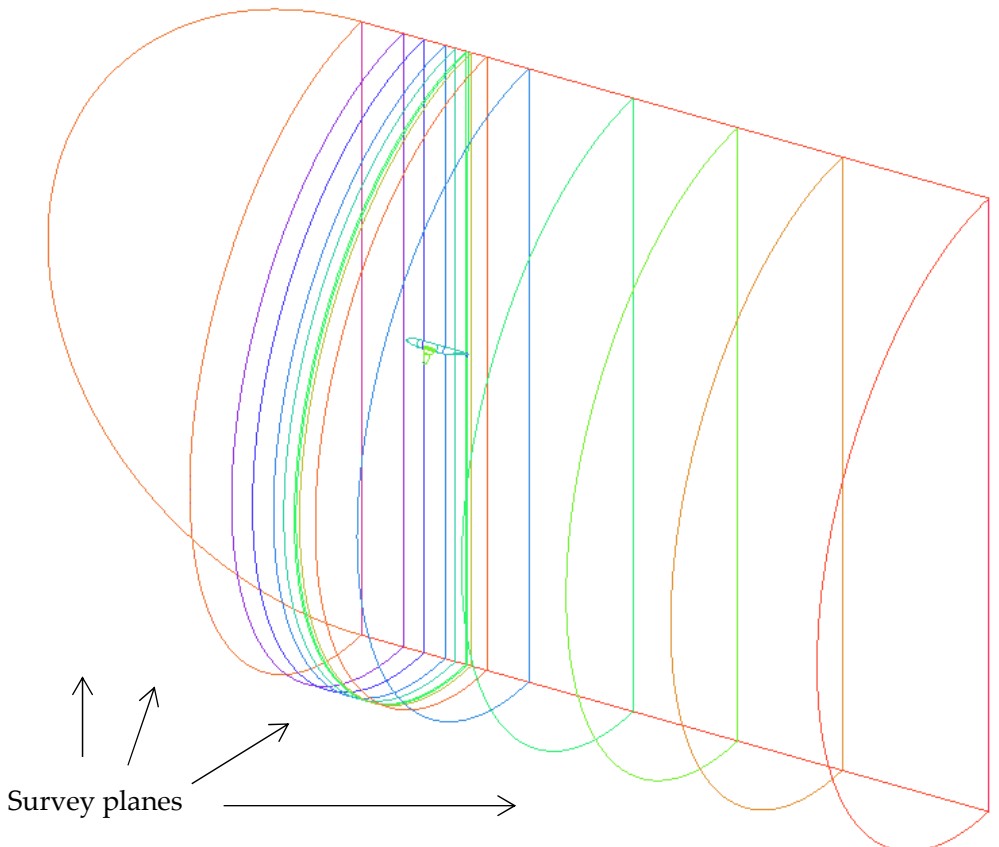

**Figure 6.** Computational domains segregated by 13 survey planes.

Meshes were generated through ICEMCFD software, trying to follow the guidelines for mesh construction of the AIAA drag prediction workshop committee [24]. Unstructured mesh and multiblock structured mesh were generated in different densities: coarse, medium, and fine. The sizes of these meshes are listed in Table 4. The unstructured mesh consisted of prismatic and tetrahedral elements. Boundary layers were captured by using 25 layers of prismatic elements, as shown in Figure 7. The multiblock structured mesh utilized O-type topology in the boundary layer region, as shown in Figure 8. For the medium density mesh, the thickness of the first-layer prism was 0.001 m, while the thickness of the first layer was 0.0006 mm (Y + approximate equals 1) for the fine mesh. The growth rate was 1.20 in the boundary layer region.

**Table 4.** Mesh size (cells).

|  | Density | Cell Number |
|---|---|---|
| Unstructured mesh | Coarse | 3.1 million |
|  | Medium | 4.1 million |
|  | Fine | 7.8 million |
| Multiblock structured mesh | Medium | 5.1 million |
|  | Fine | 8.4 million |

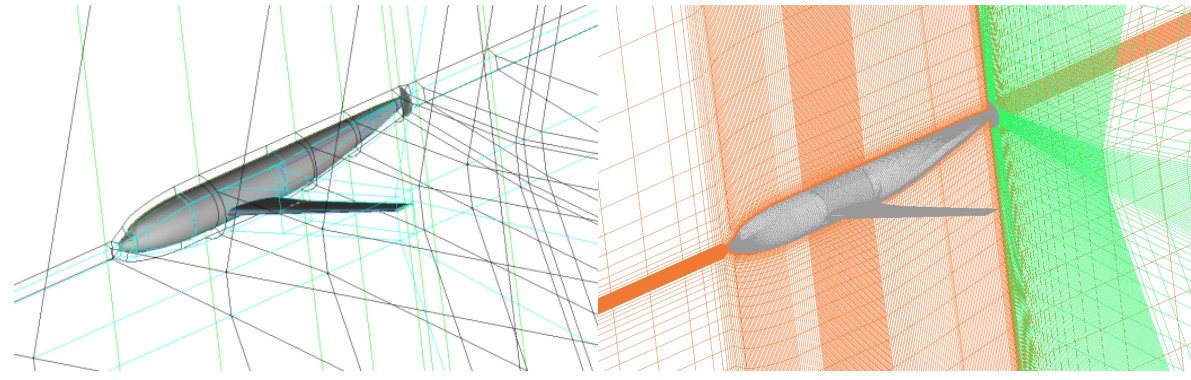

**Figure 7.** Unstructured mesh (**top left:** general view of mesh; **top right:** prism mesh over body surface; **bottom:** global view of mesh).

**Figure 8.** *Cont.*

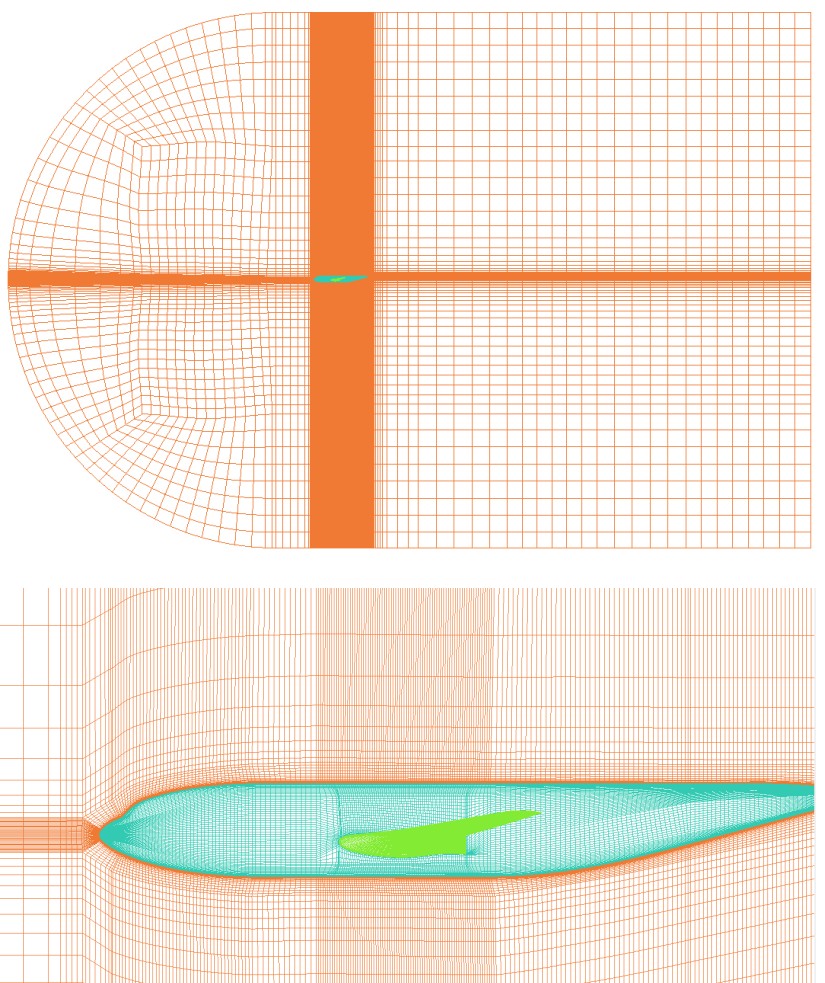

**Figure 8.** Multiblock structured mesh (**top left:** block topology; **top right:** general view of mesh; **middle:** global view of mesh; **bottom:** body and wing).

*2.4. Definition of Power Imbalance Error*

Power-based analysis is based on the conservation of mechanical energy. This study elaborates on the power imbalance error, *e*, as given in Equation (12). *e* is the difference between the power input and the power output of a domain. This provides a figure to describe how well the mechanical conservation is satisfied in this domain. In CFD simulations, the error of mechanical power imbalance can be attributed to numerical errors, which can be further broken down into modeling error, discretization error, and convergence error. In principle, numerical errors lead to the imperfection of momentum conservation, causing an imbalance in mechanical power.

$$e = \mathrm{P_{in}} - \mathrm{P_{out}} \tag{12}$$

For the entire computational domain, the power imbalance error is denoted by $e_{domain}$. The value is obtained by subtracting the $\mathrm{P_{out}}$ from $\mathrm{P_{in}}$ of the entire domain. The input and output terms are listed in Table 1. Once these power terms are assessed, the value of $e_{domain}$ can be evaluated according to Equation (13). It is noted that the shaft power, $\mathrm{P_s}$, and thrust power, $\mathrm{TV_\infty}$, are zero for the wing body configuration. For convenience purposes, the relative error is used in the following discussions. Errors are normalized with a reference power, $\mathrm{P_{ref}}$. This study uses the total power input of the entire domain as $\mathrm{P_{ref}}$: $\mathrm{DV_\infty}$ for the wing body configuration and $\mathrm{P_s}$ for BLI configuration.

$$e_{domian} = \frac{P_s + DV_\infty + +\dot{E}_{w,in} - \dot{E}_{w,out} - TV_\infty - \varnothing}{P_{ref}} \tag{13}$$

$$e_{sub} = \left( \frac{P_s + DV_\infty + \dot{E}_{w,in} - \dot{E}_{w,out} - TV_\infty - \varnothing}{P_{ref}} \right)_{subdomain} \tag{14}$$

$$e_{domain} = \sum e_{sub} \tag{15}$$

The entire computational domain contains 14 segregated subdomains. Mechanical energy conservation is satisfied for these subdomains. Therefore, it is feasible to evaluate the error of subdomains (denoted by $e_{sub}$) once all the input and output terms of the individual subdomains are obtained, as given in Equation (14). In principle, the power imbalance error of the entire computational domain, $e_{domain}$, equals to the summation of the error of all of the subdomains, $e_{sub}$, as expressed by Equation (15).

### 3. Results and Discussion

This section presents the simulation results and elaborates on the power-based analysis. The power imbalance error is analyzed, and suggestions for performing the power-based analysis are provided accordingly. The power conversion processes for the two cases of the wing body configuration and the BLI configuration are presented and discussed.

The aerodynamic forces imposed on the aircraft surface are inspected for validation purposes. The coefficients of lift and drag for the wing body configuration are assessed, and the results of different meshes are presented in Figure 9. In addition, the experiment results of the reference study are plotted as the solid line in the same figure [27]. The dash lines denote the envelope with ±3% deviation in the drag coefficient of the experiment data. Simulation results obtained in this study are on the right side of the plot of the experiment data, within the 3% deviation envelope. This indicates that the drag coefficients obtained in the simulations were higher than those of the experiment, but the difference was less than 3%. The simulation results suggest a convergence in the aircraft drag. This validation establishes a baseline for the following discussions of the power-based analysis.

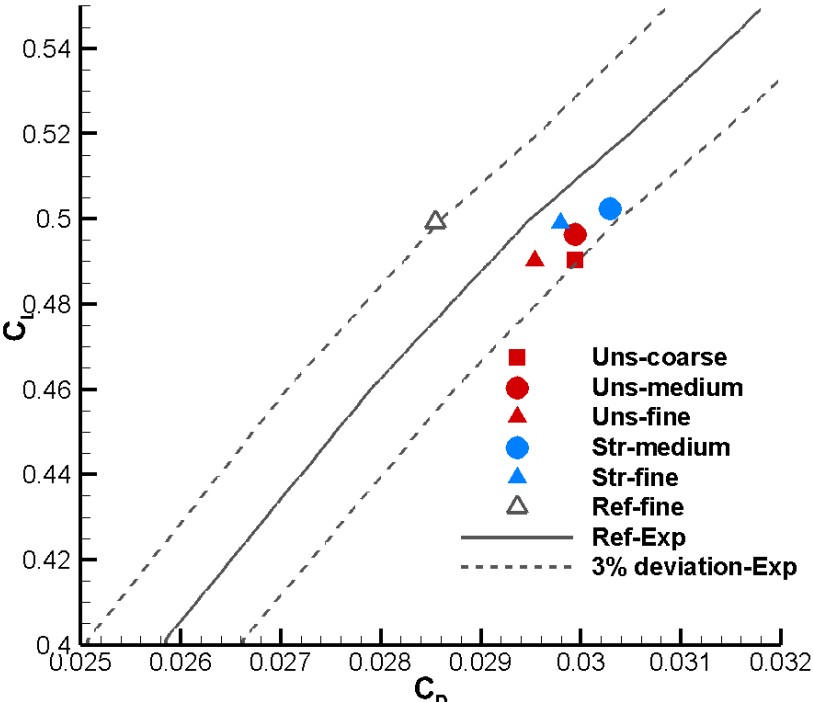

**Figure 9.** Coefficients of lift and drag for the wing body configuration.



### 3.1. The Analysis of Power Imbalance Error

Power imbalance error indicates how well mechanical energy conservation is satisfied in a computational domain. This section discusses the error in the 3D flow field of the DLR-F6 wing body configuration. Efforts were made to reduce this error in simulations.

#### 3.1.1. Mesh Size Study and Subdomain Error Study

This section presents the power imbalance error of the entire computational domain. In the first place, all the power input and output terms listed in Table 1 were evaluated. For the wing body configuration, the only power input term is the drag power, $DV_\infty$, while the output terms are the wake power at the outlet plane, $\dot{E}_{w,out}$, and the viscous dissipation, $\Phi$. Once the 3D flow field was obtained through numerical simulations, these power terms were evaluated according to their expressions, as given in Section 2.1. The power imbalance error of the entire computational domain, $e_{domain}$, was assessed according to Equation (13). The input power, $DV_\infty$, is used as the reference power to calculate the relative error of $e_{domain}$.

A mesh size study is presented in Table 5. It shows that the relative differences in drag coefficient (with respect to the experiment result) were less than 3%. The aircraft drag was independent of the mesh size and its value agreed well with the experimental results. On the other hand, the value of $e_{domain}$ ranged from 36.4% to 66.3%. Even though $e_{domain}$ tended to decrease as mesh size increased, the errors obtained in the presented 3D simulation were much higher than the 4% value obtained in the previous incompressible 2D simulations [23]. In general, the error in 3D flow simulation was far from a satisfactory level. It is critical to identify the main courses of the high value of $e_{domain}$.

**Table 5.** Mesh size study presented by the drag coefficient and the power imbalance error of the computational domain.

| Mesh | Cell Number | $C_D$ (Relative Difference) | $e_{domain}$ |
|---|---|---|---|
| Experiment [24] | - | 0.0295 | - |
| Unstructured coarse | 3.3 million | 0.0299 (1.5%) | 66.3% |
| Unstructured medium | 4.1 million | 0.0299 (1.5%) | 43.0% |
| Unstructured fine | 7.8 million | 0.0289 (−1.9%) | 38.5% |
| Structured medium | 5.1 million | 0.0303 (2.7%) | 37.5% |
| Structured fine | 8.4 million | 0.0298 (1.0) | 36.4% |

To find the major sources of power imbalance error, $e_{domain}$, this section introduces a so-called subdomain error study. As introduced in Section 2.4, it is possible to inspect the power imbalance for an individual subdomain. In this study, all the power terms listed in Table 1 were assessed for the individual subdomains. With these inputs, the errors of the power imbalance for the individual subdomains, $e_{sub}$, were computed according to Equation (14). For demonstration purposes, the results of the unstructured medium mesh are presented. The errors were plotted at the corresponding X location of the survey planes, as shown in Figure 10. The suberror denotes the error of an individual subdomain, while the bulk-error is defined as the accumulation of the suberrors from the first subdomain to the subdomain at the X location. This bulk error depicts how the error increases along the X axis. It is noted that the bulk error at the outlet plane simply equals $e_{domain}$.

This subdomain error study clearly showed that the error in the subdomain of the wing (denoted by $e_{wing}$) made the major contribution to $e_{domian}$. The $e_{wing}$ value was 26.8%, while $e_{domain}$ was 43.0%. The contributions of the other subdomains were significantly low: The suberror for the frontal body was less than 5%. The suberror in the downstream region was less than 0.5%. The error associated with the wing, $e_{wing}$, was mainly responsible for the high errors of $e_{domain}$. This study suggests that efforts should be made to reduce the suberror associated with the wing, $e_{wing}$.

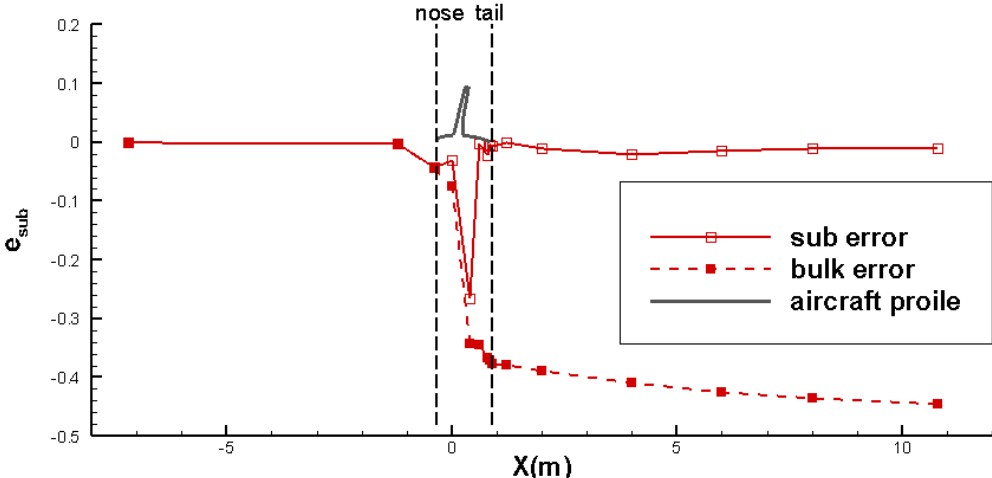

**Figure 10.** Subdomain error study of wing body configuration (unstructured medium mesh).

3.1.2. Impact Factors of Power Imbalance Error

This section presents possible impact factors to reduce $e_{domain}$. The influences of the mesh type, discretization scheme, and turbulence model are examined. Considering that the error associated with the wing, $e_{wing}$, makes the major contribution to $e_{domain}$, the subdomain of the wing is highlighted.

As introduced in Section 2.3, both structured mesh and unstructured mesh were generated. The mesh sizes of the two were close: 4.1 million for the unstructured medium mesh and 5.1 million for the structured medium mesh. It was straightforward to perform the subdomain error study with their simulation results, as presented in Figure 11. It showed the same error contributions for the two types of mesh: high-value error existed in the subdomain of the wing, which was mainly responsible for the high values of $e_{domain}$. The structured mesh indeed reduced the error in the wake region (less than 1%). Nevertheless, the error associated with the wing, $e_{wing}$ (35.4%), was higher than that of the unstructured mesh (26.8%). The errors of $e_{domain}$ and $e_{wing}$ are summarized in Table 6. Even if the structured mesh obtained relatively low values of $e_{domain}$, it did not offer a better result in terms of $e_{wing}$, the major error source. This study adopts the unstructured medium mesh as the reference mesh in the following discussions.

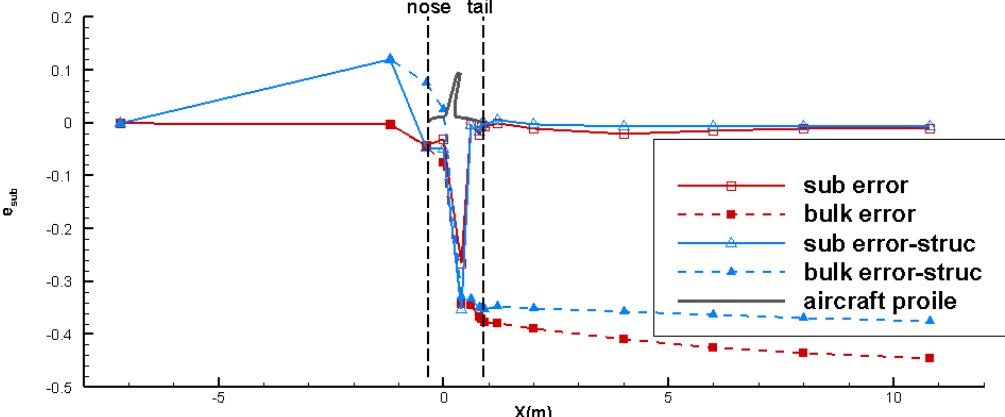

**Figure 11.** Subdomain error study for unstructured mesh (red lines) and multiblock structured mesh (blue lines).

**Table 6.** Power imbalance error for unstructured mesh and multiblock structured mesh.

| Mesh Type | $e_{domain}$ | $e_{wing}$ |
|---|---|---|
| Unstructured mesh | 43.0% | 26.8% |
| Multiblock structured mesh | 36.4% | 25.6% |

In CFD simulations, discretization methods interpolate conservative independent variables. On the other hand, interpolation causes discretization error as an important source of numerical errors. This study tested two methods for comparison: the second-order upwind and the third-order MUSCL methods. The former method is widely used, and the latter method provides higher-order precision. A first-order discretization method is not discussed due to the low order precision. A subdomain error study was performed to check the impact of the discretization methods on the reference mesh. The results are listed in Table 7. The results showed that the reduction in $e_{wing}$ due to the third-order MUSCL method was about 4.5%. Furthermore, the high-order discretization method reduced $e_{wing}$ without increasing the error of other subdomains. Therefore, the high-order discretization method is suggested for power-based analysis.

**Table 7.** Power imbalance error for the discretization methods.

| Discretization Methods | $e_{domain}$ | $e_{wing}$ |
|---|---|---|
| Second-order upwind | 43.0% | 26.8% |
| Third-order MUSCL | 38.5% | 22.3% |

A turbulent model introduces complexity to simulations, and the impact of turbulent models was examined. This study examined three typical models: the S-A model, the standard k-ε model, and the k-ω SST model. The simulations employed the same reference mesh (unstructured medium mesh) and third-order MUSCL method discretization to eliminate the effects of the other impact factors. The results of the subdomain error study are listed in Table 8. It showed that the absolute value of $e_{domain}$ ranged from 38.5% to 40.9%. The two-equation k-ω SST model led to the minimum level of power imbalance error.

**Table 8.** Power imbalance error for three turbulent models.

| Turbulent Model | $e_{domain}$ | $e_{wing}$ |
|---|---|---|
| S-A model | 40.8% | 23.0% |
| standard k-ε model | 40.9% | 22.3% |
| k-ω SST model | 38.5% | 22.3% |

As discussed in Section 3.1.1, the mesh size study showed that an increase in mesh size tended to decrease $e_{domain}$. It might be beneficial to increase the mesh density in the subdomain of the wing. Based on the reference mesh (Unstructured medium mesh), the mesh density was increased universally over the wing surface. The total mesh size was increased accordingly from 4.1 million to about 8.0 million. This refinement of the mesh led to the reduction in $e_{domain}$ from 38.5% to 36.6%, while $e_{wing}$ was reduced from 22.3% to 22.0%. The increase in the mesh density of the wing reduced the power imbalance error by 2% at the cost of the doubled mesh size. Compared with the results of the unstructured fine mesh (8.2 million), the universal increase in mesh density over the wing surface did not provide an extra benefit.

A more efficient way to increase the mesh density was examined through the mesh adaptation technique. Based on the reference mesh, the mesh density was increased in the regions with the high gradient of wake power, $\dot{E}_w$, After mesh adaptation, the total cell number increased from 4 million to about 8 million. The results showed that the $e_{domain}$ was reduced from 38.5% to 34.1% and $e_{wing}$ was reduced from 22.3% to 17.6%. The mesh density

was mainly increased in the outer region of the boundary layer of the wing surface and in the vicinity of the trailing edge. This effort led to a notable improvement compared with the results of the unstructured fine mesh. The mesh adaptation obtained the best result for $e_{domain}$ (34.1%) in this study.

In summary, the study of impact factors showed that a multiblock structured mesh obtained less total power imbalance error than the unstructured mesh, especially in the wake region. The high-order discretization method was beneficial in reducing the error. The impact of the turbulent model was marginal. Increasing the mesh density in the outer region of the wing boundary layer could reduce the power imbalance error. In the best scenario, $e_{domain}$ was 34.1% and $e_{wing}$ was 17.6% for the wing body configuration. These results are used in the following discussions.

### 3.2. Power Conversion of Wing Body Configuration

This section aims to illustrate the mechanical power conversion process of the wing body configuration. All power terms listed in Table 1 in the flow field were assessed. For the wing body configuration, the wake energy inflow (rate), $\dot{E}_{w,in}$, for the entire computational domain was zero due to the free stream flow inflow. Shaft power, $P_s$, was not involved in this case. The only mechanical power input was the drag power, $DV_\infty$, and it was used as the reference power for the normalization of power terms in this case.

The output power terms of the wing body configuration include the wake energy flow (rate), $\dot{E}_w$, and viscous dissipation, $\Phi$. $\dot{E}_w$ was assessed at the survey planes and the outlet plane. The integrand of the wake power, $\dot{E}_w$, was visualized at the survey planes, as shown in Figure 12. This figure visualizes the wake power passing through the aircraft and in the downstream flow. Crossing the aircraft, the high-intensity wake power over the body surface is highlighted. This region was thicker than the boundary layer of the body, and it was associated with the wake of the wing root. The original design of the DLR-F6 geometry suffers flow separation issues at the body–wing junction [29], as shown on the bottom right of Figure 12. High-intensity $\dot{E}_w$ was also observed in the vicinity of the wing tip, denoting the high wake power of the wing tip vortices. In the downstream region, the intensity of the wake power gradually reduced, and the area of high wake power separated into two parts: the one corresponding to the body wake and the one for wing tip vortices.

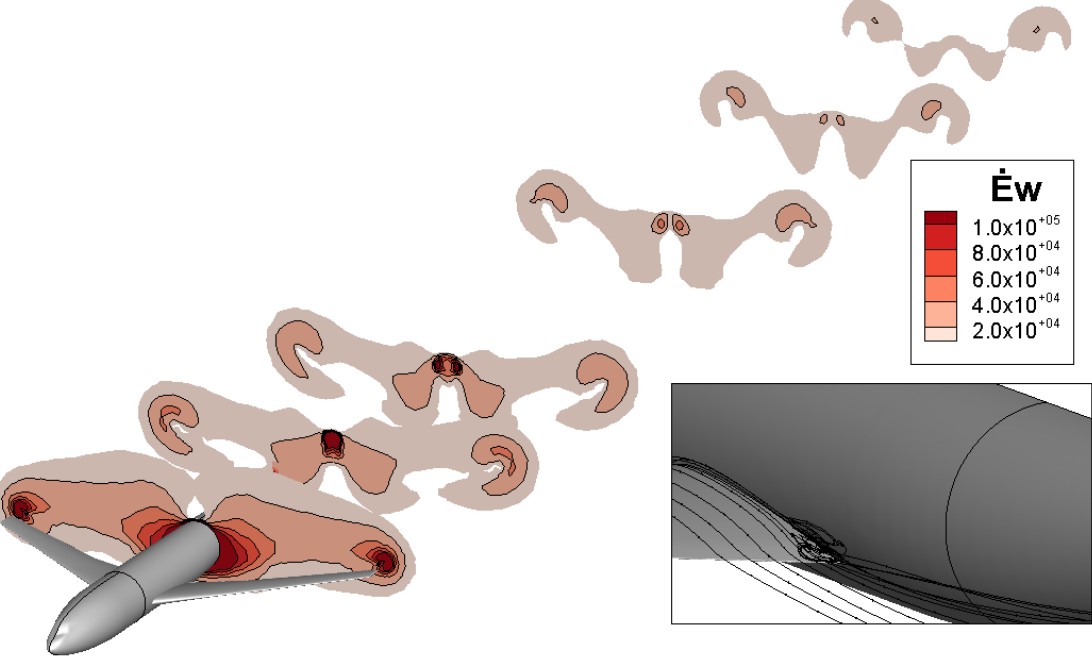

**Figure 12.** Contours of the $\dot{E}_w$ integrand around the wing body configuration.

Viscous dissipation, $\Phi$, is the sink term of power. As elaborated in Section 2.1, it exists in the viscous region (boundary layer and wake) where the velocity gradient exists. The top plot in Figure 13 illustrates the integrand of the viscous dissipation in the flow field. A high intensity of $\Phi$ was observed in the region over the airframe surface and in the downstream wake. In the downstream wake, the intensity of $\Phi$ in the body wake was higher than that of the wing wake. Compared with the wing wake, the body wake might be less stable, and its mechanical energy dissipated at a faster pace.

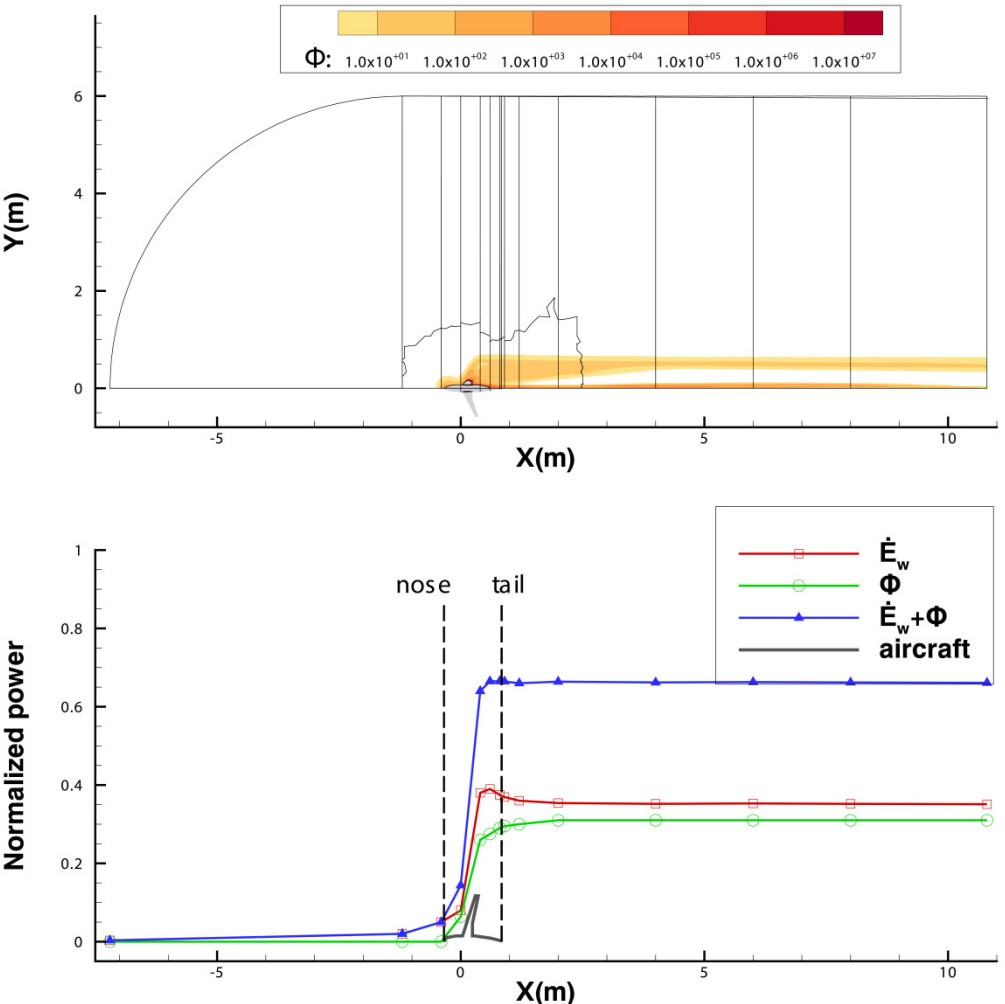

**Figure 13.** The intensity of viscous dissipation and the power conversion process of the wing body configuration.

This study visualized the change in power terms by placing their values at the corresponding X coordinates, as shown at the bottom of Figure 13. This plot illustrates how the power conversion process occurred in the flow field: drag power, $DV_{\infty}$, was transferred into wake power, $\dot{E}_w$, as well as viscous dissipation, $\Phi$.

The wake power, $\dot{E}_w$, increased quickly after the nose, reaching the peak value (37% of the total power input) after the wing. After that, it dropped quickly after the tail and decreased slowly downstream. The abrupt increase in $\dot{E}_w$ was the result of mechanical power accumulation due to the boundary layer, the separated flow in the body–wing junction, and the wing tip vortices. The mild decrease in $\dot{E}_w$ downstream of the fuselage corresponded to the dissipation of the aircraft wake. At the outlet plane, $\dot{E}_w$ remained 35% of the total power input, indicating a substantial amount of mechanical power in the aircraft wake, as shown in Figure 12.

Viscous dissipation, $\Phi$, remained almost zero before the aircraft nose, mildly increased along the frontal fuselage, and quickly increased through the wing region. In general, it kept increasing from the nose to the tail. At the X locus of the tail, $\Phi$ took about 31% of the total power input, mainly corresponding to the mechanical power loss associated with the boundary layer of the aircraft surface, while in the downstream region, the increase in $\Phi$ was less than 2%, indicating a relatively slow process of dissipation in the downstream wake. It was noted that the power input was about 34.1% higher than the summation of the output power at the outlet plane. The difference was the error of the mechanical power imbalance, as discussed in Section 3.1. The error in this 3D simulation was significantly higher than the 4% value in the previous study of 2D turbulent flow simulation [23].

### 3.3. Power Conversion of Boundary Layer Ingestion Configuration

Various sources in the previous study on BLI showed that the value of power saving due to BLI is in the range of 3~10%. On the other hand, the error of mechanical power imbalance is about 30%. The precision of power-based analysis in 3D simulation is not adequate to study the power saving of BLI. This section limits the discussion to the aspect of the power conversion process of the aircraft using BLI.

The BLI configuration combines the wing body configuration with a BLI propulsor. This study employed a wake-filling actuator disc model with a nonuniform pressure jump to mimic an ideal BLI propulsor [13,31]. This model re-energized the ingested boundary layer flow into the state of free stream so that the mechanical energy of the downstream wake and jet was minimized. The actuator disc model utilized a UDF function, which increased the total pressure of the ingested flow into a constant total pressure, $P_{t,const}$. In this case the $P_{t,const}$ value was 46,000 Pascal, higher than the free stream total pressure. In this manner, the thrust, T, generated by the actuator disc model was higher than the airframe drag, D. The UDF function enabled a nonuniform pressure jump $\Delta P$, which was simply equal to $P_{t,const}$-$P_{t,upstream}$. $P_{t,upstream}$ refers to the total pressure of the upstream flow. The thrust of this actuator disc model, T, was obtained as the surface integral of $\Delta P$ over the disc plane, as given in Equation (16). The T value was 116 N, so the $TV_\infty$ value of 30,232 watts was obtained. It was noted that $TV_\infty$ was considered a power output term, while the actual power input of the propulsor was shaft power, $P_s$, which was evaluated using Equation (7). In this BLI case, the $P_s$ value was 54,793 watts and was used as the reference value for normalizing the power terms.

$$T = \oint_{prop} \Delta P dS = \oint_{prop} \left(P_{t,const} - P_{t,upstream}\right) dS \tag{16}$$

Figure 14 shows the total pressure distribution at the symmetry plane in the vicinity of the actuator disc model. The boundary layer of the body is denoted by the contours of low-valued total pressure over the body surface. It was noted that the boundary layer over the bottom surface was thicker than that over the top surface. This was attributed to the non-axisymmetric pressure diffusion of the aft body shape. The total pressure was not uniform in the region behind the actuator disc model. Therefore, the actuator disc model was not capable of refilling the total pressure in an ideal manner. This could be attributed to the highly distorted flow ingested by the actuator disc. This study suggests that the aft body should be axisymmetric to best utilize the BLI benefit.

Aircraft drag, D, is the drag force imposed on the wing and body surface. The drag coefficient was 0.03290 in the BLI configuration. That was about 8% higher than the value in the wing body configuration. The increase in drag was due to the suction of the actuator disc model imposed on the aft body. The drag power, $DV_\infty$, is obtained by multiplying D by $V_\infty$ as a power input of the flow field.

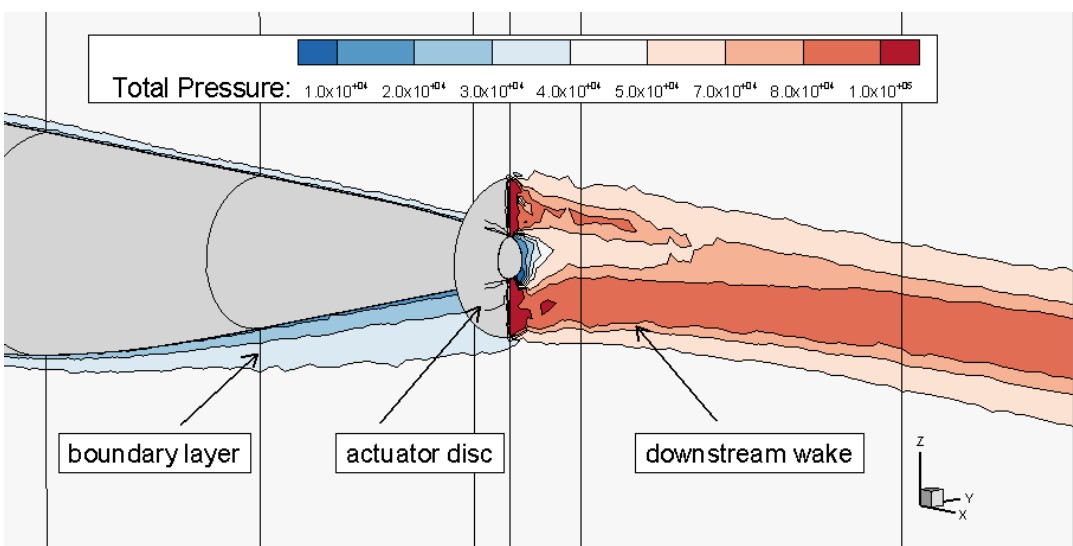

**Figure 14.** Total pressure contours in the vicinity of the wake-filling actuator disc model.

As discussed, the wake energy flow, $\dot{E}_w$, was assessed as the surface integral over the survey planes and the outlet plane. The integrand of $\dot{E}_w$ is illustrated in Figure 15. Compared with the pattern in the case of the wing body configuration. The pattern in the BLI case is similar: the wake power accumulated as the flow passed through the aircraft and gradually reduced in the downstream region. Compared with the wing body configuration, a notable difference for the BLI configuration was the high intensity of $\dot{E}_w$ in the downstream flow of the actuator disc. The downstream wake power was increased rather than decreased in this configuration. This was associated with the excessive high-momentum addition of the actuator disc: the thrust not only balanced the body drag but also overcame the wing drag.

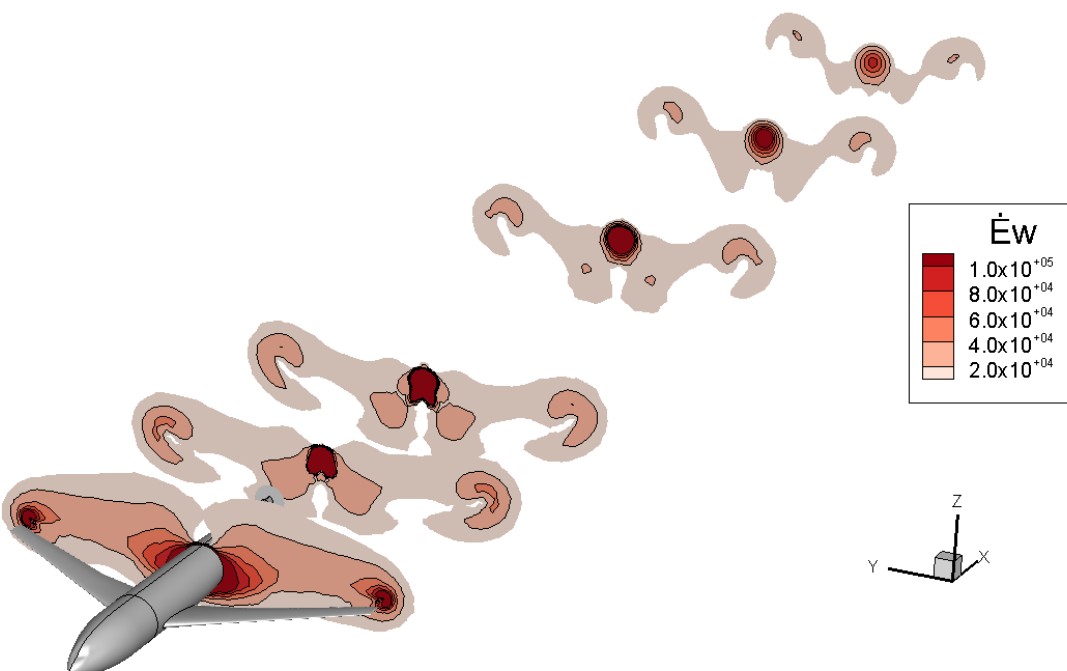

**Figure 15.** Contours of $\dot{E}_w$ integrand around the BLI configuration.

The top picture of Figure 16 illustrates the intensity of $\Phi$ in the flow field. The high intensity of $\Phi$ is highlighted around the airframe surface and in the aircraft's downstream

region. Once all of the power inputs and outputs terms in the flow field were calculated, they were normalized and plotted at the corresponding X coordinates, as shown in Figure 16. This figure illustrates the power conversion process of the BLI configuration with a propulsor. It shares a similar pattern to that of the wing body configuration.

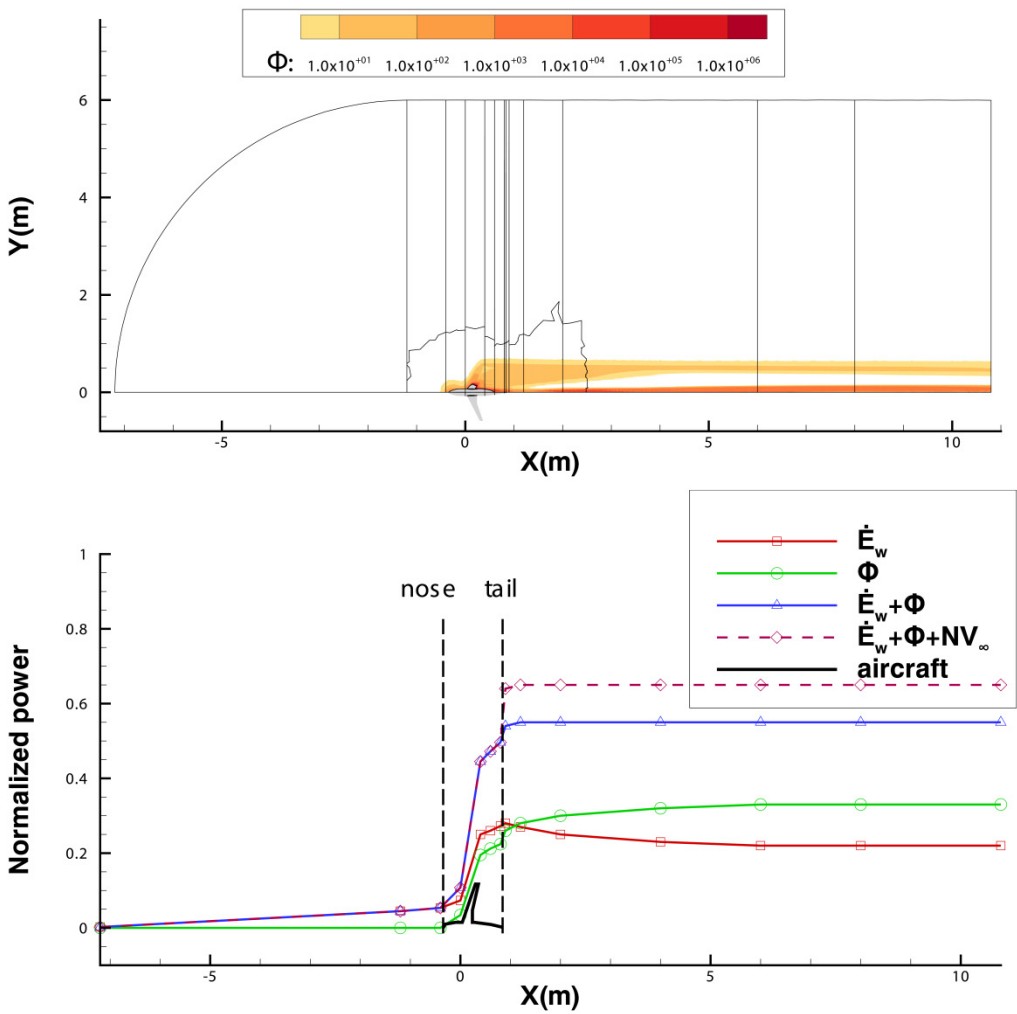

**Figure 16.** The intensity of viscous dissipation and the power conversion process of the BLI configuration.

The bottom plot in Figure 16 illustrates the power conversion process of the BLI configuration. The shaft power, $P_s$, as the only power input in the flow field, was implanted in the flow field by the propulsor. $P_s$ was converted into viscous dissipation, $\Phi$, and wake power at the outlet plane, $\dot{E}_{w,out}$, as well as the net force power, $NV_\infty$.

The viscous dissipation, $\Phi$, increased along the stream and shared a similar pattern as in the wing body configuration. It increased quickly over the aircraft surface, taking about 28% of the total power input at the body tail. Then, $\Phi$ increased by 5% in the downstream region. Besides the viscous dissipation of the aircraft wake, the increase in $\Phi$ in the downstream region was associated with the dissipation of the jet behind the actuator disc model.

The wake power, $\dot{E}_w$, increases across the airframe and decreases in the downstream reflow. At the outlet, $\dot{E}_w$ took 22% of the total input power. This indicated a notable amount of mechanical power in the downstream flow mixed with the jet and wake. This study shows the limitations of a single wake-filling actuator disc model: It can refill the body wake, but it is not able to reduce the wing wake. On the other hand, the momentum excess of the actuator disc model was required to balance the momentum deficit of the wing.

Therefore, the actuator disc model in this BLI configuration was not capable of reducing the downstream mechanical power to zero.

For the BLI configuration, the thrust power, $TV_\infty$, and drag power, $DV_\infty$, mutually cancelled each other, leaving a finite value (10% of the total input power) of the net force power, $NV_\infty$, as a power output term. $NV_\infty$ was used to accelerate the aircraft. This was different from the wing body configuration, in which $NV_\infty$ was zero to maintain the thrust/drag equilibrium. In the plot of power conversion, the total power input, $P_s$, was not identical to the summation of the outputs in the flow field. The difference was about 35%, denoting the power imbalance error for the BLI case.

A subdomain error study was performed to process the simulation results of the BLI configuration, as presented in Figure 17. It was clear that the error associated with the wing was 19.3%, which was significantly higher than the errors from the other subdomains, while the error of the BLI propulsor was 8.2%. This study shows that the error associated with the wing as well as the propulsor was mainly responsible for the high values of power imbalance error. The error of the mechanical power imbalance was larger than the typical values of power saving due to BLI. So far, the precision of power-based analysis in 3D simulation is not adequate to analyze the power saving of BLI. This poses challenges for future work.

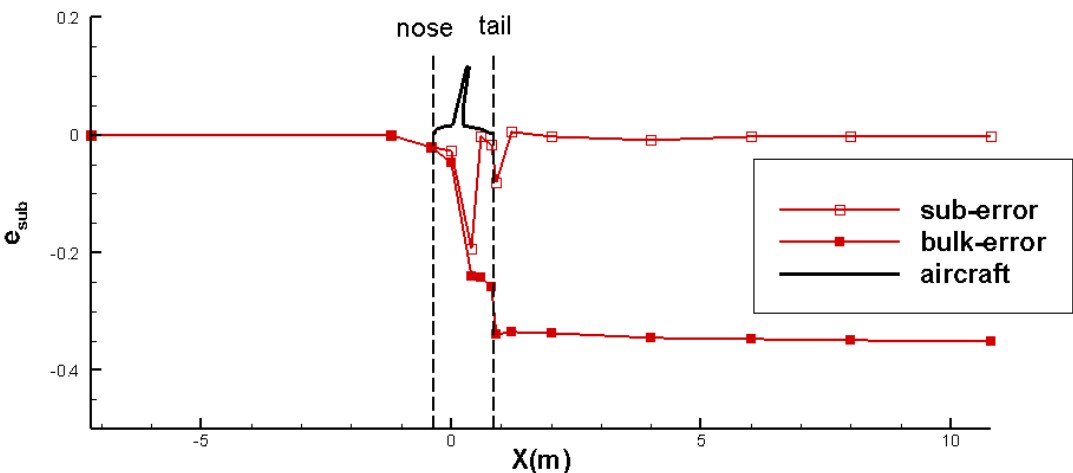

**Figure 17.** Subdomain error study of BLI configuration.

### 4. Conclusions and Recommendations

A power-based analysis was performed to study the flow field over a transonic transport aircraft through 3D compressible Reynolds-averaged Navier–Stokes simulations. The error of mechanical power imbalance in the segregated computational domain was examined. The power conversion process of the wing body configuration and the boundary layer ingestion configuration were studied and presented.

Power-based analysis was capable of illustrating the process of power conversion in the 3D flow field, clearly visualizing the change in power associated with the aircraft components. For the wing body configuration, the input power, $DV_\infty$, was converted into viscous dissipation and wake power at the outlet plane. Viscous dissipation mainly occurred in the boundary layer over the aircraft surface, taking up 31% of the total power input. The mechanical power of the aircraft wake took about 35% of the total power input at the outlet. For the BLI configuration, the input power, $P_s$, provided by the actuator disc model was converted into viscous dissipation and wake power at the outlet plane as well as net force power. The viscous dissipation of the boundary layer over the aircraft surface was about 28% of the total power input. The wake power of the downstream flow took about 22% of the total power input at the outlet plane. About 10% of the total power input was in the form of net force power that accelerated the aircraft.

The simulation results showed that the power imbalance error of the entire computational domain was 34.1% for the DLR-F6 wing body configuration, even if the difference in the drag coefficient was less than 3% compared to the experimental results. This study shows that the convergence in aircraft drag does not necessarily lead to a small power imbalance error. The error in 3D simulation is much higher than that in 2D simulations. The high power imbalance error is mainly associated with the wing.

Efforts were made in this work to reduce the power imbalance error. High-order discretization methods and an increasing mesh density in the outer region of the wing boundary layer and in the vicinity of the trailing edge were beneficial for reducing the error. However, the high value of error limits the precision of power-based analysis applied to 3D RANS simulations. Attempts to increase the mesh size to a higher order of magnitude were kept for future work due to limited computational resources in this study. Moreover, alternative simulation methods could be considered for power-based analysis in future work.

**Author Contributions:** Conceptualization, writing—original draft preparation, P.L.; Resources, writing—review and D.L.; writing—review and editing, L.M. All authors have read and agreed to the published version of the manuscript.

**Funding:** This research received no external funding.

**Institutional Review Board Statement:** Not applicable.

**Informed Consent Statement:** Not applicable.

**Conflicts of Interest:** The authors declare no conflict of interest.

**Abbreviations**

| | |
|---|---|
| BLI | Boundary Layer Ingestion |
| CFD | Computational Fluid Dynamics |
| DLR | Deutsches Zen-trum für Luft- und Raum-fahrt (German Aerospace Center) |
| RANS | Reynolds-Averaged Navier–Stokes |
| MAC | Mean Aerodynamic Chord |

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
