# Peer review of "A Power Based Analysis for a Transonic Transport Aircraft Configuration through 3D RANS Simulations"

_applsci, doi:10.3390/app122010194_

Round 1
Reviewer 1 Report (New Reviewer)
REFERENCES MUST BE INCREASED. HIGHLIGHTS AND GRAPHICAL ABSTRACT MUST BE PROVIDED. THIS IS A GOOD QUALITY WORK.
Author Response
11 relevant references have been added to the revised manuscript;
The highlights of this work are provided in the introduction;
According to the reviewer's suggestion, a graphical abstract has been attached to the manuscript on the final page.
Thanks for the kind words of the reviewer.
Reviewer 2 Report (New Reviewer)
The present study presents an analysis of transonic transport aircraft configuration through 3D RANS simulations. This study employs a segregated 3D computational domain to track the change of power terms in the flow field so that the conversion process of mechanical power is studied and visualized. Paper fulfills the scope and standard of Journal requirements. Authors need to consider the following suggestions to improve the manuscript.
1) Abstract is quite confusing and needs to rewrite.
2) Introduction is well organized; nonetheless, it would be needed to guide better the reader to understand why this research is needed. In this sense, a particular objective should be defined considering previous ideas. It should be underlined which is the paper novelty.
3) There is a lot of abbreviation used in the work hence authors are suggested to provide a nomenclature table in the introduction section which will help readers.
4) Need to add some more literature at least 10 more.
5) Authors can use comparison tables for providing a better understanding to readers.
Author Response
1) Thanks for the suggestion, abstract is rewritten as suggested by the reviewer.
2) The introduction is updated by highlighting the novelty of this work, as suggested by the reviewer.
3) Abbreviation table has been added in the introduction section as suggested by the reviewers.
4) 11 relevant references have been added to the revised manuscript.
5) We believed that 4 tables elaborating simulation results provide sufficient information to readers, therefore, we don’t add extra tables.
Reviewer 3 Report (New Reviewer)
Accept
Author Response
Thanks for the support of the reviewer!
Reviewer 4 Report (New Reviewer)
The topic is very interesting and the aim of the paper and how it was implemented are clear, and the provided information’s informative. However, the innovation is not outstanding enough. It is necessary to highlight the innovation to increase the novelty of the article. Therefore, there should be a systematic review of the literature. The authors should point out the shortcomings in other literature and highlight the differences between this paper and other literature as Table.
Author Response
The introduction is updated as suggested by the reviewer. The shortcomings in other literature are pointed out and the novelty of this work is highlighted in the revised manuscript. Suggestions of the reviewer are appreciated.
Reviewer 5 Report (New Reviewer)
The authors presented a well-conducted numerical simulation of a transonic transport aircraft. However, the author should elaborate on the nolvoties of the work: what are the contributions of this work to the scientific community?
Author Response
The introduction is updated as suggested by the reviewer. The novelty of this work is highlighted in the revised manuscript. Suggestions of the reviewer are appreciated.
Reviewer 6 Report (New Reviewer)
There are minor editing errors in the paper that should be corrected by the authors:
Different labeling of the quantities E ̇ and È, respectively (Table 1., row 117 and the following text),
different labeling of the quantity V∞, V∞
Some of the figures have low resolution. I give the authors the option to consider replacing them with better quality images.
Author Response
1. The labeling of the quantity of E ̇ has been corrected as suggested by the reviewer.
2. The labeling issue of V∞ has been corrected in the manuscript.
3. Figures 4, 13, and 16 have been updated to have higher resolution.
Thanks for the corrections of the reviewer.
This manuscript is a resubmission of an earlier submission. The following is a list of the peer review reports and author responses from that submission.
Round 1
Reviewer 2 Report
A segregated 3D computational domain to trace the change of power terms in the more realistic flow filed for a transonic transport aircraft. Comments are as below.
1. The application using BLI is not clear. The authors should explain why a wake filling actuator disc was used.
2. The authors' statement: "The simulation of wing body configuration shows that the error of power imbalance is 34.1%. Under this circumstance, it is not sensible to discuss the BLI benefit quantitatively" It is not acceptable for journal submission.
3. The technical writing is poor.
citation formation: e.g., "Baskaran, Corte [13]: should be "Baskaran et al. [13]"
line 88: an error "e" establish the equilibrium of mechanical power; line 214-215: "e" is defined as the difference between the power input and the power output of a computational domain.
Lines 436-441; lines 470-474: the statements are not clear.
Equation 1 and Equation 12 are the same
Lines 66, 68: DLR-F6, F6
typos: Control volume (line 118); Theintegrand (line 124); 0.75 Mach number (lines 143-144); the standard k-"e" model, and Mentor’s k-"w" SST model (line 188); the t reference power for normalization (line 288), and so on.